# Biology-Guided Prototype Booster: Enhancing Latent Representations of Foundation Models for Gene Expression Prediction

## Abstract

Spatial transcriptomics (ST) is a cutting-edge technology that enables the measurement of gene expression while preserving spatial context and generating detailed tissue images. However, ST technology remains time-consuming and costly. The ability to predict ST gene markers of cancer from histology-grade H&E-stained tissue images is opening new horizons for precision and personalised pathology. Despite the success of foundation models in generating general-purpose embeddings of H&E-images, these representations are not optimized for gene expression prediction and lack task-specific adaptability. To address this limitation, we introduce Biology-Guided Prototype Booster (BP-Booster), leveraging biological prior knowledge to guide the construction and training of learnable prototypes for embedding reconstruction, thereby improving gene expression prediction. We demonstrate superior performance of BP-Booster across datasets, various cancer tissue types and different ST platforms. We also show that BP-Booster can flexibly integrate various foundation models to enhance their task-specific representations, enhancing explainability and applicability in clinically relevant tasks like predicting cancer biomarkers. Code will be released upon acceptance.

## 1 Introduction

Spatial transcriptomics (ST) technologies, such as Visium, overcome the inherent lack of molecular information in H&E-stained images by enabling spatially resolved gene expression profiling and linking tissue morphology to molecular function Ståhl et al. (2016). It reveals cellular heterogeneity across tissue regions and has been globally recognized as the "Method of the Year" Marx (2021), highlighting its significant potential in both basic and translational research. However, current ST protocols are costly and labor-intensive, posing challenges for large-scale and routine clinical deployment. To address the challenges of acquiring ST data, several approaches have been proposed to predict gene expression profiles directly from H&E-stained images using deep learning techniques. For instance, ST-Net He et al. (2020) used a DenseNet-121 backbone pre-trained on ImageNet to extract image features, followed by a fully connected layer to predict gene expression. HisToGene Pang et al. (2021) incorporated spatial context by linking patch features to their tissue locations, while TRIPLEX Chung et al. (2024) further improved performance through multi-scale feature integration. However, because these models rely solely on image information, their embeddings often fail to align well with gene expression. To mitigate this gap, BLEEP Xie et al. (2023) employed contrastive learning to embed both modalities into a shared space and retrieve expression profiles via nearest-neighbor search.

In recent years, leveraging pathology foundation models (e.g., UNI Chen et al. (2024) and CONCH Lu et al. (2024)) to extract image embeddings from H&E-stained slides and predict gene expression has emerged as an active research direction. This line of work benefits from the strong representational power of foundation models but still yields suboptimal results. Foundation models are primarily trained with self-supervised learning on large-scale pathology image collections, enabling their embeddings to capture broad morphological patterns. However, these embeddings often contain redundant information and are not explicitly optimized for downstream tasks such as gene expression prediction. To mitigate this limitation, Stem Zhu et al. (2025) employed joint distribution modeling to align gene expression and image embeddings in a shared latent space, but its effectiveness

depends heavily on large training datasets. As an alternative, dimensionality reduction techniques such as Principal Component Analysis (PCA) reduce high-dimensional data by projecting it onto directions of maximum variance, thereby preserving the most informative structures. Experiments on the HEST benchmark Jaume et al. (2024) show that applying PCA to image embeddings from pathology foundation models improves gene expression prediction, suggesting that the original embeddings may contain task-irrelevant components that hinder prediction performance. However, as PCA prioritizes high-variance directions, it may also discard lower-variance but more biologically meaningful features. Therefore, developing an effective strategy to directly reconstruct these embeddings by incorporating task-specific biological information has great potential to improve predictive performance.

Motivated by these observations, we propose a lightweight module called the Biology-Guided Prototype Booster (BP-Booster), which incorporates biological prior knowledge to guide the construction and learning of task-specific prototypes, thereby refining image embeddings from foundation models. By reconstructing the embeddings through BP-Booster, task-irrelevant information is removed, while biologically meaningful semantic features are preserved, leading to improved gene expression prediction. At the core of BP-Booster is a prototype-guided cross-attention module that facilitates interaction between image embeddings and learnable prototypes. In addition, the joint supervised reconstruction–regression objective in BP-Booster refines the image embeddings by preserving essential visual information while simultaneously injecting biologically meaningful signals derived from selected gene expression profiles. Furthermore, we introduce a prototype initialization strategy that includes both random initialization and gene program-guided initialization, offering flexibility to integrate either purely data-driven signals or biologically informed priors during training, thereby establishing a strong foundation for learning task-specific prototypes tailored to gene expression prediction. It is worth noting that for gene program-based initialization, we construct gene programs by combining spatially variable genes (SVGs) and biological pathways. SVGs capture spatial variations in gene expression across tissues, while pathways group genes with coordinated biological functions. Prototypes initialized in this way are not only biologically meaningful but also better aligned to interact with image embeddings.

The main contributions of this work can be summarized as follows:

- We propose BP-Booster, a lightweight, biology-guided prototype-based cross-attention module that directly reconstructs image embeddings from foundation models by incorporating biological prior knowledge;

- We apply a joint reconstruction-regression optimization strategy to enable refined image embeddings by BP-Booster, enabling BP-Booster to refine image embeddings that preserve essential visual information while injecting biologically meaningful signals to improve gene expression prediction;

- We propose a prototype initialization strategy, where gene program-based initialization imparts biological semantics to prototypes and improves their interaction with image embeddings;

- Extensive experiments demonstrate that BP-Booster achieves superior performance and strong generalizability across diverse datasets and foundation models, while also proving effective for gene expression signature prediction.

## 2 METHODOLOGY

### 2.1 PROBLEM FORMULATION AND METHOD OVERVIEW

#### 2.1.1 PROBLEM FORMULATION.

Given a set of H&E image patches $I_N \in \mathbb{R}^{H \times W \times 3}$ and the corresponding gene expression profiles $G \in \mathbb{R}^{N \times C}$, where $H \times W$ denotes the spatial resolution of each patch, $C$ is the number of genes, and $N$ is the number of patches. We further define $G^{\text{HVG}} \in \mathbb{R}^{N \times K_H}$ and $G^{\text{SVG}} \in \mathbb{R}^{N \times K_S}$ as subsets of $G$ corresponding to highly variable genes (HVGs) and spatially variable genes (SVGs), respectively, where $K_H$ and $K_S$ are the number of selected genes in each category.

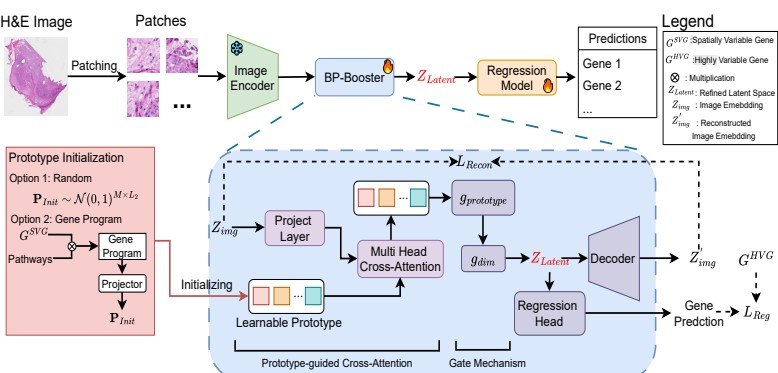

Figure 1: **The pipeline of our BP-Booster.** The H&E image is split into patches, and each patch is encoded into $Z_{\text{img}}$ using a frozen pathology foundation model. Learnable prototypes are initialized through a prototype initialization strategy and combined with $Z_{\text{img}}$ via cross-attention and gating to produce refined embeddings $Z_{\text{Latent}}$. These embeddings are optimized with reconstruction and regression losses guided by $Z_{\text{img}}$ and $G^{\text{HVG}}$, and subsequently used to train a regression model for gene expression prediction. The pseudo-code for **BP-Booster** is provided in the Appendix A.3.

The gene expression prediction task is typically formulated as a regression problem. Existing methods often leverage a pathology foundation model to extract image embeddings $Z_{\text{img}} \in \mathbb{R}^{N \times L_1}$, where $L_1$ is the embedding dimension. A regression model (e.g., Ridge regression) is then trained on $Z_{\text{img}}$ to predict $G$. While our method follows this general paradigm, it introduces a key innovation: we propose BP-Booster to transforms $Z_{\text{img}}$ into a refined latent embedding $Z_{\text{latent}} \in \mathbb{R}^{N \times L_2}$, where $L_2$ denotes the dimensionality of the latent space. This transformation enhances the robustness and task alignment of the representations, ultimately improving predictive performance.

### 2.1.2 METHOD OVERVIEW.

An overview of the BP-Booster pipeline is presented in Figure 1. During training, the model leverages the triplet $I_N$, $G^{\text{HVG}}$, $G^{\text{SVG}}$. First, a foundation model extracts image embeddings $Z_{\text{img}}$ from $I_N$. Then, a set of learnable prototypes $P \in \mathbb{R}^{M \times L_2}$ in BP-Booster is initialized by the prototype initialization strategy (either randomly or based on gene programs derived from $G^{\text{SVG}}$). BP-Booster learns to project the image embedding $Z_{\text{img}}$ into a refined latent space $Z_{\text{latent}}$ through interactions between image features and learnable prototypes, guided by $G^{\text{HVG}}$. After training, $Z_{\text{latent}}$ is combined with $G$ to train a regression model for gene expression prediction. At inference time, only the H&E image patches are used. They are processed through the foundation model and the trained BP-Booster to obtain $Z_{\text{latent}}$, which is then used by the regression model to predict gene expression.

## 2.2 BP-BOOSTER

Reconstructing $Z_{\text{img}}$ using biological priors presents a core challenge: how to leverage one modality (gene expression) to guide the latent space refinement of another (image embeddings). Inspired by the Q-Former Li et al. (2023) designed to extract task-relevant visual features for text generation, we propose BP-Booster, a novel biologically driven refinement module. BP-Booster consists of four key components: a prototype-guided cross-attention module, gating mechanisms for prototypes and latent dimensions, a decoder, and a regression head.

### 2.2.1 PROTOTYPE-GUIDED CROSS-ATTENTION MODULE.

The core of BP-Booster is a lightweight prototype-guided cross-attention module composed of: a projection layer, a learnable prototype set $P$ and a multi-head cross-attention layer. Since the dimension of $Z_{\text{img}}$ can vary across foundation models, we first apply a projection layer to align it

with the prototype dimension $L_2$. The multi-head cross-attention module then computes prototype-aware representations $Z_P \in \mathbb{R}^{N \times M \times L_2}$ by taking $\mathbf{Proj}(Z_{\text{img}})$ and $P$ as inputs, as follows:

$$
\begin{aligned}
Z_P &= \text{MultiHead}(P, \mathbf{Proj}(Z_{\text{img}}), \mathbf{Proj}(Z_{\text{img}})) \\
&= \text{Concat}(H_1, \ldots, H_h)W^O
\end{aligned}
\tag{1}
$$

$$
\begin{aligned}
Where\ H_i &= \text{Attention}(P, \mathbf{Proj}(Z_{\text{img}}), \mathbf{Proj}(Z_{\text{img}})) \\
&= \text{softmax}\left( \frac{P(\mathbf{Proj}(Z_{\text{img}}))^\top}{\sqrt{d_h}} \right) \mathbf{Proj}(Z_{\text{img}})
\end{aligned}
\tag{2}
$$

Here, $W^O \in \mathbb{R}^{L_2 \times L_2}$ is the output projection matrix, $d_h = L_2/h$ is the head dimension, and $\mathbf{Proj}(\cdot)$ is the projection layer. This design allows each prototype to attend to distinct semantic subspaces of the image embeddings, improving cross-modal alignment.

### 2.2.2 GATING MECHANISMS.

To improve representation quality and interpretability, BP-Booster incorporates two learnable gating mechanisms:

**Prototype Gate**: Each prototype is associated with a learnable scalar gate $g_{\text{prototype}} \in [0,1]^M$. The gated prototype output is:

$$
Z_P' = \sum_{m=1}^{M} \alpha_m g_{\text{prototype}}^{(m)} Z_P^{(m)}
\tag{3}
$$

where $\alpha$ denotes the attention scores from the prototype-based cross-attention layer, and $\odot$ represents element-wise multiplication. $M$ is the number of prototypes.

**Latent Dimension Gate**: A second gate $g_{\text{dim}} \in [0,1]^{L_2}$ is applied to control each latent dimension:

$$
Z_{\text{Latent}} = Z_P' \odot g_{\text{dim}}
\tag{4}
$$

These mechanisms selectively suppress uninformative prototypes and dimensions, enhancing both model compactness and task-specificity.

### 2.2.3 DECODER AND REGRESSION HEAD.

To ensure that $Z_{\text{Latent}}$ preserves essential semantic information from $Z_{\text{img}}$ while being predictive of gene expression, we introduce a decoder that reconstructs $Z_{\text{img}}$ from $Z_{\text{Latent}}$, supervised with a reconstruction loss $L_{\text{Recon}}$. A regression head that maps $Z_{\text{Latent}}$ to the top 200 HVGs from the training set, with prediction supervised by a regression loss $L_{\text{Reg}}$. Together, these losses enforce that $Z_{\text{Latent}}$ is both faithful to visual input and biologically meaningful, driving improved generalization on downstream gene expression prediction tasks.

## 2.3 PROTOTYPE INITIALIZATION STRATEGY

In BP-Booster, prototypes play a central role in transforming $Z_{\text{img}}$ into the refined latent space $Z_{\text{latent}}$. The strategy for initializing these prototypes is therefore an important design choice. To this end, we propose a prototype initialization strategy that supports two modes: random initialization and gene program-guided initialization. This strategy enables flexibility in leveraging purely data-driven or biologically informed priors during model training.

### 2.3.1 RANDOM INITIALIZATION.

The baseline initialization strategy involves sampling the learnable prototypes from a Gaussian distribution, given by:

$$
P_{\text{Init}} \sim \mathcal{N}(0,1)^{M \times L_2}.
\tag{5}
$$

To ensure that the resulting latent representation $Z_{\text{Latent}}$, constructed from these randomly initialized prototypes and the image embeddings $Z_{\text{img}}$, encodes not only essential visual features but also

biologically meaningful information related to gene expression, we employ both $G^{\text{HVG}}$ and $Z_{\text{img}}$ as supervisory signals. These guide the prototypes to capture visual-biological correspondences during training. This strategy enables the model to learn biologically relevant abstractions in a data-driven manner, without requiring prior domain knowledge for prototype initialization. However, random initialization lacks semantic grounding and makes it difficult to interpret what each prototype represents, thereby limiting the interpretability of the learned latent space. To address this limitation, we propose an alternative gene program–guided initialization strategy, driven by spatially variable genes (SVGs).

### 2.3.2 GENE PROGRAM-GUIDED INITIALIZATION.

A gene program is a set of co-expressed genes associated with a specific biological process. It is typically constructed by integrating biological pathways with data-driven co-expression patterns to identify biologically meaningful and context-specific gene modules Gerber et al. (2007). Existing gene program construction methods, such as ExpiMap Lotfollahi et al. (2023), primarily rely on filtering highly variable genes (HVGs). While HVGs capture genes with significant expression variation across samples, they often overlook the spatial organization of tissues. In contrast, spatially variable genes (SVGs) capture expression differences across distinct tissue locations, thus better reflecting spatial heterogeneity. The visualization is provided in the Appendix A.2.

Inspired by this, we propose a prototype initialization strategy driven by spatially variable genes (SVGs) and enriched with biological priors at the pathway level. This strategy enables a semantically meaningful initialization for prototypes, improving interpretability and providing a strong starting point for subsequent training. Specifically, we first collect a set of biological pathways $GP = \{GP_1, GP_2, \ldots\}$ from a pathway database such as MSigDB Liberzon et al. (2011). These pathways are filtered by intersecting them with the SVG set $G^{\text{SVG}}$, retaining only those that satisfy the condition $|G^{\text{SVG}} \cap GP| \geq 10$. Based on the filtered results, we build the pathway-SVG binary matrix $B \in \mathbb{R}^{K_S \times J}$, where $K_S$ is the number of SVGs and $J$ is the number of selected pathways. Then, the gene program matrix is defined as:

$$A = G^{\text{SVG}} B \in \mathbb{R}^{N \times J}, \tag{6}$$

Then, we apply Sigmoid activation to each gene program $A_{:,i_j}$, followed by normalization to compute the spatial weights:

$$w^{(j)} = \frac{\sigma(A_{:,i_j})}{\sum_{n=1}^{N} \sigma(A_{n,i_j})} \in \mathbb{R}^N, \tag{7}$$

where $\sigma(\cdot)$ denotes the Sigmoid activation function. Using these weights, we define the $j$-th prototype as the weighted sum of SVG expression across spatial positions:

$$p_j = \sum_{n=1}^{N} w_n^{(j)} G_n^{\text{SVG}}. \tag{8}$$

Finally, by stacking all prototypes and passing them through a projection layer, we obtain the initialized prototype matrix:

$$P_{\text{Init}} = \mathbf{Projector}([p_1^\top, \ldots, p_J^\top]) \in \mathbb{R}^{J \times L_2}, \tag{9}$$

A key advantage of this initialization strategy is that, unlike random initialization, it eliminates the need to manually specify the number of prototypes by directly leveraging gene programs as semantically meaningful prototypes (*i.e.,*, $M = J$). Moreover, since each prototype corresponds to a filtered pathway, the model gains clear biological interpretability by linking prototypes to known pathways.

### 2.4 JOINT RECONSTRUCTION-REGRESSION OPTIMIZATION

In our approach, BP-Booster is optimized using a combination of a reconstruction loss and a regression loss, defined as follows:

$$L_{\text{Recon}} = \frac{1}{N} \sum_{i=1}^{N} \left| \text{Decoder}(Z_{\text{Latent}}^{(i)}) - Z_{\text{img}}^{(i)} \right|^2 \tag{10}$$

$$L_{\text{Reg}} = \frac{1}{N} \sum_{i=1}^{N} \left| \text{RegressionHead}(Z_{\text{Latent}}^{(i)}) - G_i^{\text{HVG}} \right|^2 \tag{11}$$

The overall training objective is the sum of these two components:

$$L_{\text{Total}} = L_{\text{Recon}} + L_{\text{Reg}} \tag{12}$$

This joint optimization strategy encourages the model to learn latent representations that are not only faithful to the original image embeddings but also predictive of biologically meaningful gene expression patterns.

## 3 EXPERIMENTS

### 3.1 DATASET AND EXPERIMENT SETUP

**Dataset.** We evaluated our method on four cancer types from the HEST benchmark Jaume et al. (2024): IDC (invasive ductal carcinoma of the breast), SKCM (skin cutaneous melanoma), PRAD (prostate adenocarcinoma), and LUAD (lung adenocarcinoma). The benchmark provides a publicly available, high-quality spatial transcriptomics (ST) dataset with standardized preprocessing. For IDC, SKCM, and LUAD, data were generated using Xenium technology, comprising 4, 2, and 2 samples, respectively, and cross-validation was conducted following Jaume et al. (2024) to ensure reliable evaluation of generalization. For PRAD, data were collected using Visium technology, consisting of 23 samples, and evaluation was performed with leave-one-out as in Zhu et al. (2025).

**Gene List Selection Strategy for Training.** In our method, two biologically distinct gene sets are utilized to initialize and train the prototypes: spatially variable genes (SVGs) and highly variable genes (HVGs). SVGs are genes whose expression levels exhibit significant variation across spatial locations within a tissue, capturing spatial heterogeneity. We use Squidpy Palla et al. (2022) to identify the top 200 SVGs from the training set, which are employed for prototype initialization. HVGs represent genes with high expression variability across samples and are commonly used to capture key data structures. We select the top 200 HVGs from the training data using Scanpy Virshup et al. (2023) and use them for prototype training.

**Foundation Models.** In this study, we evaluate our method using nine foundation models across different experiments: ResNet-50 Lu et al. (2021), CTransPath Wang et al. (2022), Phikon Filiot et al. (2023), CONCH Lu et al. (2024), GigaPath Xu et al. (2024), UNI Chen et al. (2024), Virchow Vorontsov et al. (2024), Virchow 2 Zimmermann et al. (2024), and H-Optimus-0 Saillard et al. (2024).

**Implementation Details.** Our model is implemented using the PyTorch framework and executed on an NVIDIA L40 GPU (48GB memory). We use Adam optimizer to train our model for 50 epochs. The batch size is 128. The hyperparameters related to the prototype (the dimension and number) and the learning rate are adjusted based on the choice of foundation model and validation dataset. The learning rate was selected from the range [0.00001, 0.005]. The prototype dimension was chosen from {128, 256, 512}, and the number of prototypes from {16, 32, 64}. Detailed hyperparameter configurations for each experiment are provided in the Appendix A.5. For evaluation, we follow the HEST benchmark and first use the top 50 HVGs to assess performance on the IDC, SKCM, and LUAD datasets. Second, we adopt the high mean and high variability gene (HMHVG) list, as defined in Stem, to evaluate all methods on the PRAD dataset. Additionally, we conduct PAM50 gene prediction on the IDC dataset and cancer marker prediction on the SKCM dataset, demonstrating that BP-Booster is particularly effective for biologically relevant tasks. To measure gene expression prediction performance, we use Mean Squared Error (MSE) and Pearson Correlation Coefficient (PCC) as evaluation metrics.

## 3.2 EXPERIMENT RESULTS

In this section, we compare BP-Booster with existing methods, evaluate its generalization across different foundation models, and demonstrate its applicability to predicting gene expression signatures and cancer markers (results shown in A.6.1). We also present a comprehensive ablation study.

### 3.2.1 COMPARISON WITH STATE-OF-THE-ART.

We compare BP-Booster with existing methods, including ST-Net, HisToGene, TRIPLEX, Stem and BLEEP.

As shown in Table 1, BP-Booster consistently outperforms all existing methods across all evaluation metrics in both tasks, regardless of the foundation model used. This demonstrates the robustness of our approach across different cancer and tissue types on the Xenium. In addition, prototype initialization guided by gene programs (specifically those derived from SVGs) consistently yields better performance than random initialization. This indicates that spatially informed gene programs more effectively capture underlying spatial patterns, thereby facilitating more meaningful embedding reconstruction. In addition, we compare the visualizations of BP-Booster, the comparison methods, and the ground truth, which is shown in the Appendix A.6.2.

| Method | IDC (HVG) | | SKCM (HVG) | |
|---|---|---|---|---|
| | PCC ↑ | MSE ↓ | PCC ↑ | MSE ↓ |
| ST-Net | 0.4147 ±0.054 | 2.6918 ±3.43 | 0.5109 ±0.1518 | 3.914 ±5.0458 |
| HisToGene | 0.089 ±0.0204 | 2.9903 ±3.2483 | 0.2655 ±0.037 | 2.7275 ±2.304 |
| BLEEP | 0.4131 ±0.0596 | 2.6973 ±3.2692 | 0.3812 ±0.0325 | 3.2473 ±2.5385 |
| Ours_R(UNI) | 0.5791 ±0.0755 | 2.3575 ±3.3278 | 0.6337 ±0.0107 | 1.5701 ±1.6459 |
| Ours_GP(UNI) | 0.5845±0.0799 | 2.3128 ±3.2394 | 0.6239 ±0.0189 | 1.6127 ±1.723 |
| Ours_R(Virchow2) | 0.5895 ±0.0834 | 2.0904 ±3.0457 | 0.6482 ±0.013 | **1.3768 ±1.3989** |
| Ours_GP(Virchow2) | **0.5986 ±0.0844** | **2.0801 ±3.0375** | **0.6522 ±0.0171** | 1.3895 ±1.3511 |

Table 1: Comparison with state-of-the-art methods on the IDC and SKCM datasets from the HEST benchmark Jaume et al. (2024). Reported results correspond to the mean and standard deviation obtained from cross-validation. Our_R denotes BP-Booster with random initialization and Our_GP is BP-Booster with gene program-guided initialization.

To further demonstrate the generalization of our proposed method across different ST platforms and varying gene set sizes, we conduct an additional experiment on a Visium dataset (PRAD). Table 2 shows that BP-Booster significantly outperforms other methods in all settings, indicating its robustness across both platforms and prediction scales. Moreover, consistent with findings in Table 1, prototype initialization based on gene programs generally yields better performance than random initialization. However, an exception is observed when using Virchow 2, where random initialization is better than gene program-based initialization. We assume the reason could be the lower spatial resolution of the Visium with a mixture of cell types per spatial spot, where prior information from gene programs may restrict the representation ability of prototypes, limiting their ability to adapt to the complex latent space from Virchow 2.

| Method | PRAD (HMHVG) | | | |
|---|---|---|---|---|
| | PCC-10 ↑ | PCC-50 ↑ | PCC-200 ↑ | MSE ↓ |
| HisToGene* | 0.4035 | 0.3554 | 0.2235 | 1.4619 |
| BLEEP* | 0.5798 | 0.5102 | 0.3158 | 2.4754 |
| TRIPLEX* | 0.6173 | 0.4953 | 0.3601 | 1.4819 |
| Stem* | 0.6103 | 0.5315 | 0.3832 | 1.4873 |
| Ours_R(UNI) | 0.6727 | 0.6164 | 0.478 | 0.5442 |
| Ours_GP(UNI) | 0.6677 | 0.6171 | 0.4935 | 0.5384 |
| Ours_R(CONCH) | 0.6538 | 0.5983 | 0.472 | 0.5599 |
| Ours_GP(CONCH) | 0.674 | 0.6132 | 0.4946 | 0.5472 |
| Ours_R(Virchow2) | **0.7308** | **0.6807** | **0.5628** | **0.4894** |
| Ours_GP(Virchow2) | 0.7285 | 0.6759 | 0.5563 | 0.4981 |

Table 2: Comparison with state-of-the-art methods on the PRAD dataset from the HEST benchmark Jaume et al. (2024). Reported results are the mean values obtained from cross-validation. The * denotes results reported by Zhu et al. (2025).

### 3.2.2 GENERALIZATION FOR FOUNDATION MODELS.

| Method | SKCM(HVG) PCC ↑ | | | | LUAD(HVG) PCC ↑ | | | |
|---|---|---|---|---|---|---|---|---|
| | R∗ | R + PCA∗ | R + Ours_R | R + Ours_GP | R∗ | R + PCA∗ | R + Ours_R | R + Ours_GP |
| ResNet 50 | 0.4117 | 0.4822 | 0.5139 | **0.5297** | 0.4001 | 0.4917 | 0.5406 | **0.5459** |
| CTransPath | 0.4038 | 0.5196 | 0.5512 | **0.5572** | 0.4026 | 0.4985 | 0.5745 | **0.5749** |
| Phikon | 0.3684 | 0.5355 | 0.5557 | **0.5729** | 0.4224 | 0.5468 | 0.5868 | **0.5973** |
| CONCH | 0.5079 | 0.5791 | **0.5811** | 0.5642 | 0.4957 | 0.5312 | 0.5566 | **0.569** |
| GigaPath | 0.2231 | 0.5538 | 0.5704 | **0.5706** | 0.3144 | 0.5399 | 0.5594 | **0.562** |
| UNI | 0.3433 | 0.6254 | **0.6337** | 0.6239 | 0.3714 | 0.5511 | 0.5565 | **0.5795** |
| Virchow | 0.3389 | 0.6088 | 0.6464 | **0.6573** | 0.2897 | 0.5459 | 0.6067 | **0.612** |
| Virchow2 | 0.303 | 0.6174 | 0.6482 | **0.6522** | 0.3017 | 0.5605 | 0.5837 | **0.60** |
| H-Optimus-0 | 0.2778 | 0.6432 | 0.6571 | **0.6632** | 0.3143 | 0.5582 | 0.5875 | **0.5907** |
| Average | 0.3531 | 0.5739 | 0.5953 | **0.599** | 0.368 | 0.536 | 0.5725 | **0.5813** |

Table 3: Generalization performance of the proposed method across various foundation models on the SKCM and LUAD datasets from the HEST benchmark Jaume et al. (2024). Reported results are the mean values obtained from cross-validation. The ∗ denotes results reported by Jaume et al. (2024). R is the Ridge regression model.

| Ridge | PGCA module | $g_{dim}$ | $g_{prototype}$ | GP initialization | $L_{Recon}$ | $L_{Reg}$ | Virchow2 |
|---|---|---|---|---|---|---|---|
| ✓ | | | | | | | 0.303 |
| ✓ | ✓ | | | | ✓ | ✓ | 0.638 |
| ✓ | ✓ | | ✓ | | ✓ | ✓ | 0.6478 |
| ✓ | ✓ | ✓ | | | ✓ | ✓ | 0.644 |
| ✓ | ✓ | ✓ | ✓ | | ✓ | ✓ | 0.6482 |
| ✓ | ✓ | ✓ | ✓ | ✓ | ✓ | | 0.6498 |
| ✓ | ✓ | ✓ | ✓ | | | ✓ | 0.6492 |
| ✓ | ✓ | ✓ | ✓ | ✓ | ✓ | ✓ | **0.6522** |

Table 4: Ablation analysis of the proposed method on the SKCM dataset from the HEST benchmark Jaume et al. (2024). Reported results are the mean values of PCC obtained from cross-validation. PGCA module: the prototype-guided cross-attention module.

To evaluate the adaptability of BP-Booster to different foundation models, we compared it with the PCA reduction on the SKCM and LUAD datasets. As shown in Table 3, we first observe that applying PCA substantially improves the performance of foundation models, resulting in average PCC gains of 0.216 and 0.168 on the two tasks, respectively. This improvement is likely since raw image embeddings from foundation models contain a considerable amount of task-irrelevant or redundant information, which hinders effective learning by the regression model. Next, we notice that BP-Booster (with both random and gene program-based initialization) consistently outperforms PCA. For example, our method with gene program initialization increases the PCC by 0.0251 and 0.0453 on the two tasks, respectively. This indicates that BP-Booster is more effective at reconstructing a task-relevant latent space, providing a more suitable feature representation for gene expression prediction. As PCA focuses on directions of maximal variance, it potentially discards biologically informative but low-variance features. However, BP-Booster leverages biologically informed prototypes to guide latent space reconstruction. This approach improves the preservation of relevant information about gene expression while simultaneously filtering out noise and redundancy.

### 3.2.3 PREDICTION OF GENE EXPRESSION SIGNATURE

A gene expression signature is a set of genes whose expression patterns characterize specific biological states or diseases. These signatures are widely used in disease classification, prognosis, and treatment response, making their prediction with deep learning an important application. Therefore, we evaluate our method on the PAM50 Parker et al. (2009) (a breast cancer gene expression signature panel). In the IDC dataset, 12 PAM50 genes are identified. We predict these genes using Virchow 2 alone and our BP-Booster (with both random and gene program initialization). As shown in Figure 2, BP-Booster consistently outperforms Virchow 2, highlighting its effectiveness for accurate gene expression signature prediction.

### 3.3 ABLATION STUDY

In the ablation studies, we evaluate the contribution of each component within BP-Booster. The results are summarized in Table 4. We employ a Ridge regression model as the baseline for com-

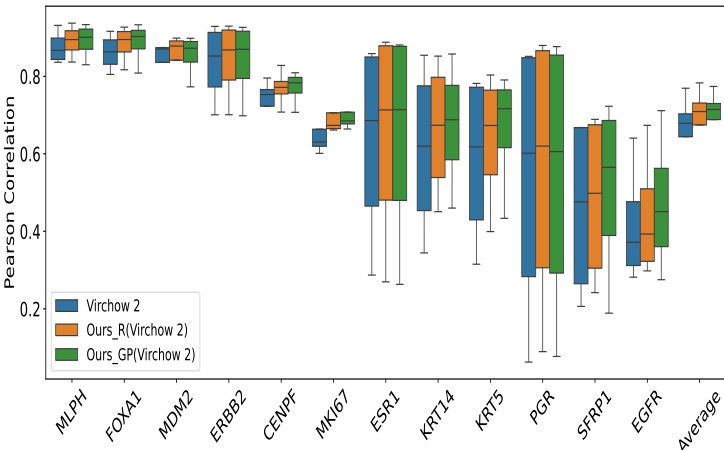

Figure 2: Comparison between the proposed method and Virchow 2 on predicting PAM50 genes in the IDC dataset.

parison. Each component is incrementally added to assess its impact on the overall performance, allowing us to isolate the effectiveness of the prototype-guided cross-attention module, gating mechanisms, gene program initialization, and two loss functions. We began by incorporating a prototype-guided cross-attention module into the baseline model, which led to significant performance improvements (PCC gain of 0.335). Next, we independently added the $g_{prototype}$ and $g_{dim}$. The results show that adding $g_{dim}$ yields greater performance gains than $g_{prototype}$. When both gating mechanisms were integrated into the prototype-guided cross-attention module, the resulting model is called BP-Booster with random initialization. It further improves performance, achieving PCC gain of 0.0102. Then, initializing the prototypes using gene programs (GP) led to additional performance gains, demonstrating the effectiveness of incorporating biological priors into the representation learning process. Finally, we evaluate the contributions of each loss function. Using the $L_{Recon}$ and $L_{Reg}$ individually produces slightly lower performance than their combined use. This likely reflects their complementary objectives: $L_{Recon}$ ensures the BP-Booster embedding retains information from the original image embedding, while $L_{Reg}$ enforces gene expression relevance in the embedding.

Additionally, we analyze three types of embeddings (i.e., image embeddings from Virchow 2, image embeddings of Virchow 2 after PCA, and embeddings from BP-Booster) using t-SNE and k-means clustering (see Appendix A.7.1). The results show that BP-Booster embeddings form clearer clusters, with tighter intra-cluster compactness and sharper inter-cluster boundaries, which may contribute to improved gene expression prediction performance. We further conduct ablation studies examining prototype configuration (number and dimensionality), gene program analysis, the influence of different gene sets used for prototype initialization and regression, and the effect of biological information injection. The results are summarized in A.7.

## 4 CONCLUSION

In this study, we propose BP-Booster, a lightweight refinement module that reconstructs latent representations from existing pathology foundation models through biologically guided prototype learning to improve gene expression prediction. Experimental results show that integrating BP-Booster not only enhances prediction performance across different ST platforms and tissue types but also strengthens the task-specific representational capacity of various foundation models, further demonstrating its applicability in the context of gene expression signatures. We expect that BP-Booster, which reconstructs foundation model image embeddings by incorporating biological knowledge, can offer new insights into leveraging foundation models and H&E images for computational pathology. In future work, we plan to extend this approach to related tasks such as cell type classification.

## REPRODUCIBILITY STATEMENT

The appendix provides hyperparameter settings and algorithmic pseudo-code to enhance reproducibility. We will release the code upon acceptance.

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

# A APPENDIX

## A.1 RELATED WORK

### A.1.1 PROTOTYPE-BASED REPRESENTATION LEARNING

A prototype refers to a representative instance that captures the shared characteristics of data points belonging to the same type or category, and is often used to encode semantic or conceptual information Song et al. (2024a). **Recently, prototype learning has gained significant attention in computational pathology and has become a widely adopted approach for H&E image analysis.** For example, AttnMISL Yao et al. (2020) and H2T Yao et al. (2020) performed k-means clustering on patch embeddings and use the resulting cluster centroids as prototypes. PANTHER Song et al. (2024a) employed a Gaussian mixture model (GMM) to summarize patches from whole-slide images (WSIs) into a set of compact morphological prototypes. MMP Song et al. (2024b) extended PANTHER by incorporating pathway-level prototypes and integrating them with morphological prototypes for downstream tasks. In contrast to these approaches, we propose to construct learnable prototypes directly guided by biological information. These prototypes are used to reconstruct image embeddings from foundation models, thereby enhancing their utility for gene expression prediction.

### A.1.2 PARAMETER-EFFICIENT FINE-TUNING

Parameter-efficient fine-tuning (PEFT) methods aim to adapt large pre-trained models while updating only a small subset of their parameters. For example, LoRA Hu et al. (2022) utilized low-rank update matrices in transformer layers, enabling efficient task adaptation without full backbone fine-tuning. In contrast, BP-Booster is not a PEFT method, which does not modify or adapt any weights of the foundation model. Instead, BP-Booster operates solely on top of frozen image embeddings, learning external parameters that refine the embedding space without altering the backbone. Furthermore, unlike PEFT approaches, BP-Booster is designed to remain effective under extremely limited sample regimes (such as spatial transcriptomics), where updating backbone parameters is impractical.

### A.1.3 COMPARE WITH OTHER TECHNIQUES

Our BP-Booster is conceptually different from domain adaptation, multi-modal alignment, and feature recalibration. First, unlike domain adaptation, which seeks to mitigate distribution shifts and learn domain-invariant features via aligning distribution approaches, our method focuses on semantically adapting visual features to molecular profiles, addressing modality-level semantic differences rather than distribution mismatch. Second, compared with multi-modal alignment methods such as contrastive learning Chen et al. (2020), which typically embed two modalities into a shared universal space for bidirectional retrieval or generation, our method performs a unidirectional refinement of image embeddings, where HVGs serve as supervised signals to guide BP-Booster in refining the embeddings rather than mapping image and gene expression features into a shared representation space. This makes the embeddings more suitable for gene expression prediction. Finally, unlike classic feature recalibration approaches such as SENet Hu et al. (2018), which compute channel weights using internal feature statistics, our method reconstructs embeddings by modelling the correlation between image features and biologically informed prototypes, providing improved interpretability.

## A.2 SPATIALLY VARIABLE GENE

Spatially variable genes (SVGs) capture expression differences across tissue locations, thereby reflecting spatial heterogeneity, as visualized in Figure 3. Incorporating SVGs makes gene programs more closely aligned with image embeddings from foundation models.

## A.3 PSEUDO CODE OF BP-BOOSTER

Algorithm 1 illustrates the training procedure of BP-Booster and how it generates the latent representation $Z_{\text{Latent}}$ from image embeddings. During training, $Z_{\text{Latent}}$ is obtained from H&E images via BP-Booster and used together with ground truth gene expression to train a regression model. At

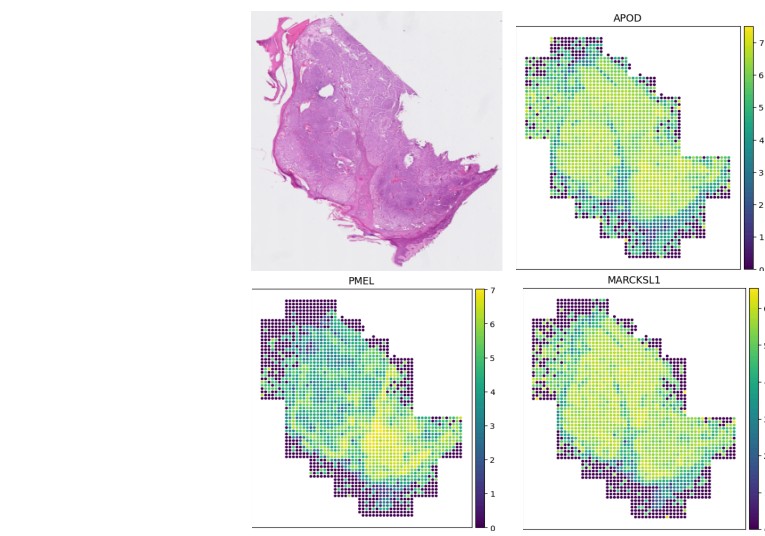

Figure 3: Visualizations of spatially variable genes (SVGs) demonstrate that their expression patterns align well with the underlying spatial organization of the tissue.

test time, the trained BP-Booster generates $Z_{\mathrm{Latent}}$ from test H&E images, which is then fed into the trained regression model to predict gene expression.

## A.4 DATA PREPROCESSING

We follow the HEST benchmarkJaume et al. (2024) to preprocess data across four cancer types. Highly variable genes (HVGs) and spatially variable genes (SVGs) are selected only from training data. For datasets with multiple training samples, we retain genes that are commonly identified as HVGs or SVGs across all samples.

## A.5 HYPERPARAMETER SETTINGS

We detail the hyperparameters used in our experiments, including learning rate, prototype dimensions, and prototype counts (for random initialization only).

### A.5.1 LEARNING RATE

Different learning rates are used based on the dataset and initialization strategy (*i.e.,*random and gene program-guided initialization):

- IDC (random): 0.01;
- LUAD (both): 0.001;
- IDC (gene program): 0.0001;
- SKCM (random, gene program): 0.0005;
- PRAD (random): 0.0001;
- PRAD (gene program): 0.00001.

### A.5.2 PROTOTYPE DIMENSION

The prototype dimension is set to 256 for most foundation models. Exceptions: ResNet-50 (512), CONCH and UNI (128).

---

**Algorithm 1** A pipeline of BP-Booster.

---

**BP-Booster:** Prototypes $P$, project layer $\text{Proj}$, prototype gate $g_{\text{prototype}}$, latent dimension gate $g_{\text{dim}}$, decoder $D$, regression head $R$, prototype-guided cross-attention layer $\text{MultiHead}$

**Input:** Foundation models $FM$, patches from train H&E images $I_N$, highly variable genes $G^{\text{HVG}}$, spatially variable genes $G^{\text{SVG}}$

**Output:** Reconstructed image embeddings $Z_{\text{Latent}}$

**Prototype Initialization:**
**if** random initialization **then**
    $P \sim \mathcal{N}(0, 1)$
**else if** gene program-guided initialization **then**
    $GP = \{GP_1, GP_2, \ldots\}$                     $\triangleright$ GP denotes gene program
    **if** $|G^{\text{SVG}} \cap GP| \geq 10$ **then**
        $A = G^{\text{SVG}} B$                $\triangleright$ $B$ denotes pathway-SVG binary matrix
        $w^{(j)} = \frac{\sigma(A_{:,i_j})}{\sum_{n=1}^{N} \sigma(A_{n,i_j})}$     $\triangleright$ $\sigma(\cdot)$ denotes the Sigmoid activation function
    **end if**
    $P = \text{Proj}([p_1^{\top}, \ldots, p_J^{\top}])$             $\triangleright$ $p_j = \sum_{n=1}^{N} w_n^{(j)} G_n^{\text{SVG}}$
**end if**

**Training:**
**for** $epoch \in 1, 2, \ldots, n$ **do**

$$Z_{\text{Latent}} = g_{\text{dim}} \odot \Big[ \sum \alpha \cdot g_{\text{prototype}}\cdot$$
$$\text{MultiHead}(P, \text{Proj}(Z_{\text{img}}), \text{Proj}(Z_{\text{img}})) \Big]$$

                                $\triangleright$ $\alpha$ denotes the attention score

    where $Z_{\text{img}} = FM(I_N)$,
    Compute $L_{\text{Recon}}$ using $Z_{\text{img}}$ and $D(Z_{\text{Latent}})$ to train **BP-Booster**
    Compute $L_{\text{Reg}}$ using $G^{\text{HVG}}$ and $R(Z_{\text{Latent}})$ to train **BP-Booster**
**end for**

---

### A.5.3 NUMBER OF PROTOTYPES

The gene program-guided strategy automatically determines the number of prototypes. For random initialization:

- Most of the foundation models: 64;
- CONCH and UNI: 32.

### A.6 ADDITIONAL EXPERIMENTS

### A.6.1 PREDICTION OF CANCER MARKERS

To further assess whether biological relevance is the key driver behind BP-Booster's effectiveness (i.e., comparing random versus gene-program initialization), we evaluated cancer markers prediction on the SKCM dataset, where the selected genes are closely tied to tumour phenotype and microenvironmental states. The results, presented in Table 5, align with expectations: models whose representations are biologically grounded (UNI, Virchow2) benefit substantially from GP initialization. In contrast, CONCH, whose representations are dominated by text-aligned histopathological semantics rather than biological pathway structure, performs better with random initialization.

### A.6.2 VISUALIZATION RESULTS

In this section, we present visualization results in Figure 4, comparing BP-Booster, comparison methods, and the ground truth for cancer-related gene expression predictions.

| Method | Random Initialization | Gene Program Initialization |
|---|---|---|
| | PCC ↑ | PCC ↑ |
| UNI | 0.861 | **0.8663** |
| CONCH | **0.8686** | 0.8583 |
| Virchow 2 | 0.8342 | **0.8518** |

Table 5: Comparison with methods on the SKCM cancer marker prediction task.

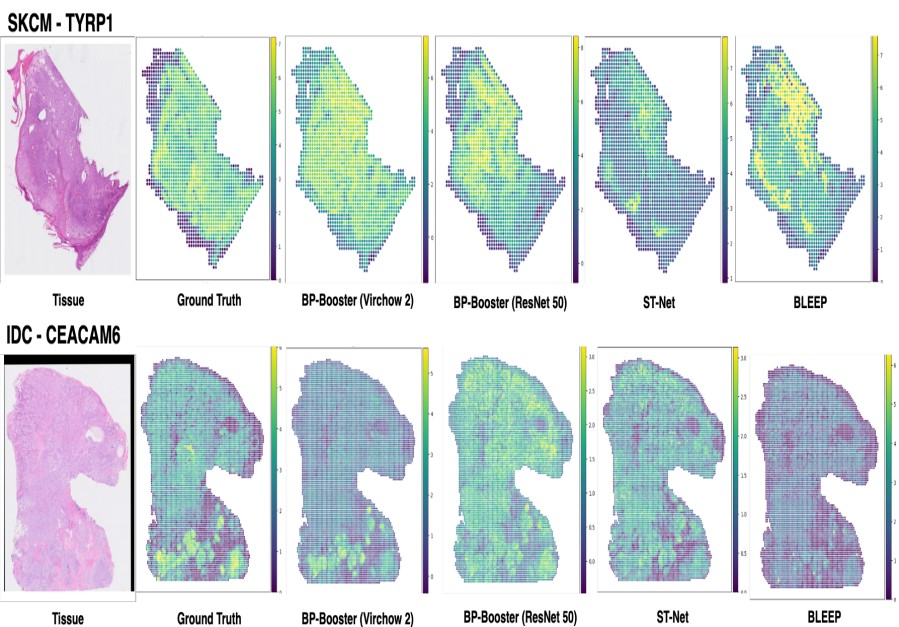

Figure 4: Visualization of cancer-related gene expression predictions (*e.g.*TYRP1 and CEACAM6) from different methods on the IDC and SKCM datasets.

## A.7 ABLATION STUDIES

### A.7.1 ANALYZE EMBEDDINGS

We employed t-SNE and k-means clustering to assess the representation power of different embeddings on the SKCM dataset. As shown in Figure 5, raw embeddings from Virchow 2 were highly dispersed with substantial cluster overlap, leading to blurred boundaries. PCA reduced this overlap and produced more compact clusters, though separation remained limited. In contrast, embeddings from our method yielded clearly separated clusters with higher intra-cluster density and sharper boundaries, demonstrating superior representational capability. These improvements suggest potential benefits for downstream gene expression prediction, as further validated in our main experiments.

### A.7.2 DIFFERENT GENE SETS FOR PROTOTYPE INITIALIZATION AND SUPERVISED REGRESSION

The rationale for using two distinct gene sets (SVGs and HVGs) stems from the different roles of prototype initialization and supervised regression. Using SVGs to construct gene programs because they provide prototypes with spatially coherent and biologically meaningful priors, aligning well with the morphological patterns captured in H & E image embeddings. In contrast, not all HVGs carry such spatial information. Moreover, HVGs are used for supervised regression because their high expression variability generates clearer gradients, offering stronger supervision signals. To validate this design, we conducted additional experiments by flipping the roles of SVGs and HVGs on the SKCM dataset for HVGs and cancer markers prediction (which are shown in Table 6). The results show that using SVGs for prototype initialization and HVGs for regression consistently outperforms the flipped configuration.

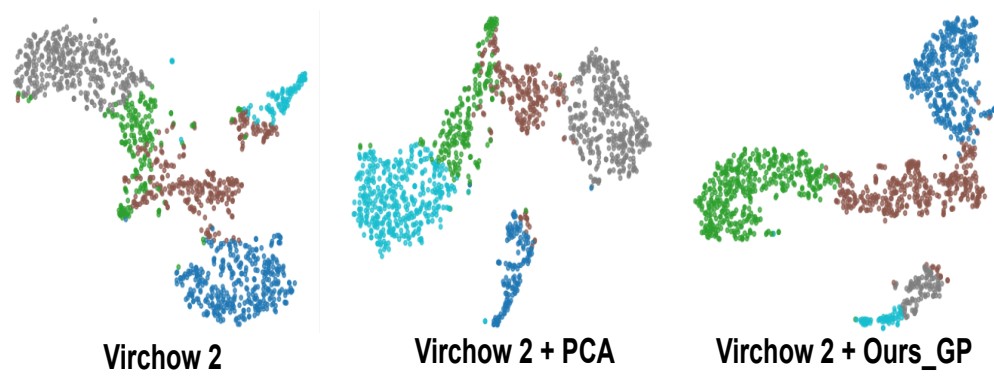

Figure 5: T-SNE visualization with k-means clustering of different embeddings on the SKCM task.

| Method | Top 50 HVGs PCC ↑ | Cancer Markers PCC ↑ |
|---|---|---|
| SVGs init + HVGs regression | **0.6562** | **0.8518** |
| HVGs init + SVGs regression | 0.6510 | 0.8342 |

Table 6: Comparison of two different strategies across two prediction tasks on the SKCM dataset. The foundation model is Virchow 2.

### A.7.3 ANALYZE PROTOTYPE

**Number of Prototype.** We perform ablation studies to investigate how model performance and the number of prototypes is affected by varying the threshold used in gene program initialization, as shown in Table 7. When the threshold is set to 1, meaning a gene program is activated if it shares at least one SVG with the selected genes, a total of 64 gene programs is included. However, this leads to decreased performance compared to a threshold of 10. The likely reason is that some activated gene programs contain only a single SVG, thus introducing noise during initialization rather than providing useful prior knowledge. Conversely, when we increase the threshold to 15, only two gene programs remain after filtering. In this case, the prior information becomes overly restrictive, providing insufficient biological guidance and resulting in a slight decline in model performance. These observations demonstrate that the threshold plays a critical role in balancing the richness and noise level of prior knowledge incorporated into prototype initialization.

| Threshold | Number of Prototypes | Top 50 HVGs PCC ↑ |
|---|---|---|
| 1 | 64 | 0.6471 |
| 10 | 12 | **0.6562** |
| 15 | 2 | 0.6548 |

Table 7: Ablation study on prototype number sensitivity by varying the threshold in gene program initialization on the SKCM dataset (Virchow 2 as foundation model).

**Dimension of Prototype.** We also conducted ablation studies on the prototype dimensionality for the top 50 HVGs prediction task, as shown in Table 8. The results indicate that different cancer types exhibit different performance trends under Virchow 2 when the prototype dimension changes. For selecting the default prototype dimension, we consider all datasets for each foundation model to determine a balanced setting.

### A.7.4 ANALYZE GENE PROGRAM

For different disease types, the gene programs differ. We extract gene programs list for SKCM and LUAD datasets, which is shown in Table 9. The table shows that different cancer type has different numbers and types of pathways selected for prototype initialization. Lung cancer is dominated

| Dimension of Prototype | SKCM PCC ↑ | LUAD PCC ↑ |
|---|---|---|
| 128 | 0.6634 | 0.5959 |
| 256 | 0.6562 | 0.6004 |
| 512 | 0.6293 | 0.5794 |

Table 8: Performance comparison of different prototype dimensions on the SKCM and LUAD datasets for the top 50 HVGs prediction task (using Virchow 2).

by classical oncogenic signalling, including EGFR, ALK, KRAS–RAF–MEK, and AKT–mTOR, leading to enrichment of genes in RTK activation and cell-cycle progression Tj (2004); Drosten & Barbacid (2020). In contrast, skin cancer, especially melanoma, shows more of stemness and developmental pathways such as Wnt/LEF1 and SHH, together with RAF activation, reflecting more genes in dedifferentiated and MAPK-driven profiles Davies et al. (2002).

| SKCM | LUAD |
|---|---|
| P53 DN.V1 DOWN | HOXA9 DN.V1 UP |
| LEF1 UP.V1 UP | MEK UP.V1 UP |
| SNF5 DN.V1 UP | RPS14 DN.V1 UP |
| ESC V6.5 UP EARLY.V1 DOWN | EGFR UP.V1 UP |
| CAMP UP.V1 DN | ALK DN.V1 UP |
| RAF UP.V1 DOWN | AKT UP MTOR DN.V1 DOWN |
| P53 DN.V1 UP | |
| NFE2L2.V2 | |
| BMI1 DN MEL18 DN.V1 UP | |
| BMI1 DN.V1 UP | |
| STK33 SKM UP | |
| GCNP SHH UP EARLY.V1 DOWN | |

Table 9: Gene program lists for the SKCM and LUAD datasets derived from MSigDB pathways.

### A.7.5 ANALYZE BIOLOGICAL INFORMATION INJECTION

To isolate the contribution of biological information (including both biological priors and biological supervision), we replace BP-Booster with a simple MLP-based gated autoencoder. This model uses the same decoder and regression head as BP-Booster, and its encoder contains the same number of parameters as PGCA. We conducted an ablation study on the SKCM dataset for the top 50 HVGs prediction task. As shown in Table 10, BP-Booster consistently outperforms the gated autoencoder, demonstrating that the performance gains stem from the incorporation of biological information rather than from additional model parameters.

| Method | Top 50 HVGs PCC ↑ |
|---|---|
| Gated Autoencoder + reconstruction loss | 0.5973 |
| Gated Autoencoder + reconstruction + regression loss | 0.6448 |
| BP-Booster | **0.6562** |

Table 10: Comparison among a simple MLP-based autoencoder (with and without biological supervision) and BP-Booster on the top 50 HVGs prediction task in the SKCM dataset (Virchow 2 as foundation model).

### A.8 THE USE OF LARGE LANGUAGE MODELS (LLMS)

The large language models (LLMs) are used solely to improve grammar, spelling and wording.

