# OpenReview forum: "Biology-Guided Prototype Booster: Enhancing Latent Representations of Foundation Models for Gene Expression Prediction"
_ICLR.cc/2026/Conference — Submitted to ICLR 2026_

### Official Review · Reviewer_N92h · 2025-10-31

**Soundness:** 3
**Presentation:** 2
**Contribution:** 2
**Rating:** 2
**Confidence:** 3

**Summary:**

This paper proposes BP-Booster (Biology-Guided Prototype Booster), a lightweight refinement module that enhances the latent representations of pathology foundation models for the task of gene expression prediction from H&E-stained histology images. The manuscript points out the limitation that foundation model embeddings, though powerful, are not tailored for gene expression prediction and often contain task-irrelevant information.
Experiments are conducted on multiple cancer types and ST platforms, showing consistent improvements over existing methods such as ST-Net, HisToGene, BLEEP, TRIPLEX, and Stem. BP-Booster also demonstrates robustness across different foundation models and applicability to predicting specific gene expression signatures.

**Strengths:**

- The paper makes a clear statement that general-purpose pathology embeddings are sub-optimal for tasks requiring alignment with gene expression data and the practice to solve the mis-alignment is to some extent of great importance to the community.
- The biologically-guided prototype learning with cross-attention and gating is an intuitive yet effective way to refine embeddings without re-training the foundation model.
- Extensive comparisons with SOTA methods on multiple datasets and ST technologies are provided and the results are satisfactory.

**Weaknesses:**

- The authors seem to misuse the tex commands of \citep{} and \citet{}, leading to incorrect citation formats. Please consider to correct these commands in a revised version.
- While the paper motivates BP‑Booster well from a computational pathology perspective, conceptually the approach can be interpreted as a variant of feature transformation / domain adaptation. The module takes source H&E patch embeddings from a frozen foundation model and maps them into a target latent space aligned with gene expression outputs using prior knowledge. This idea is related to existing techniques in domain adaptation, multimodal alignment, and feature recalibration. However, the manuscript fails to discuss these related works in the manuscript.

**Questions:**

- For the readers' information, have you tried fine-tuning the foundation model encoder jointly with BP-Booster? Would this further improve results or lead to overfitting on small datasets?
- How sensitive is BP-Booster to the number of prototypes and prototype dimension? Is there a risk of overparameterization in smaller datasets?
- My deepest concern towards this method is the positioning. Can you elaborate your method's contributions under the context of the concepts mentioned in the section of weaknesses? If not relevant, why the practice is unique? If yes, why existing methods cannot solve the issue?

---

> ### Author Response · Authors · 2025-11-24
> **Q4-1: While the paper motivates BP‑Booster well from a computational pathology perspective, conceptually the approach can be interpreted as a variant of feature transformation / domain adaptation. However, the manuscript fails to discuss these related works in the manuscript.**
>
> A4-1: Our approach is conceptually different from domain adaptation, multi-modal alignment, and feature recalibration. First, unlike domain adaptation, which seeks to mitigate distribution shifts and learn domain-invariant features via align distribution approaches, our method focuses on semantically adapting visual features to molecular profiles, addressing modality-level semantic differences rather than distribution mismatch. Second, compared with multi-modal alignment methods such as contrastive learning (Chen etal. 2020), which typically embed two modalities into a shared universal space for bidirectional retrieval or generation, our method performs a unidirectional refinement of image embeddings, where HVGs serve as supervised signals to guide BP-Booster in refining the embeddings rather than mapping image and gene expression features into a shared representation space. This makes the embeddings more suitable for gene expression prediction. Finally, unlike classic feature recalibration approaches such as SENet (Hu etal. 2018), which compute channel weights using internal feature statistics, our method reconstructs embeddings by modelling the correlation between image features and biologically informed prototypes, providing improved interpretability.  We will include these comparisons in the revised version.
>
> Reference:
>
> Chen, T., Kornblith, S., Norouzi, M., & Hinton, G. (2020, November). A simple framework for contrastive learning of visual representations. In International conference on machine learning (pp. 1597-1607). PmLR.
>
> Hu, J., Shen, L., & Sun, G. (2018). Squeeze-and-excitation networks. In Proceedings of the IEEE conference on computer vision and pattern recognition (pp. 7132-7141).

---

> > ### Author Response · Authors · 2025-11-24
> > **Q4-2: For the readers' information, have you tried fine-tuning the foundation model encoder jointly with BP-Booster? Would this further improve results or lead to overfitting on small datasets?**
> >
> > A4-2: We did not perform joint fine-tuning of the foundation model and BP-Booster because this setup is fundamentally misaligned with the problem setting we address. Our focus is on extremely data-limited scenarios (compared to the number of samples used in training or fine-tuning foundation models),  in which even PEFT methods may overfit. Conceptually, BP-Booster is designed to operate on frozen embeddings, and its reconstruction–regression objective assumes a stable embedding space. Jointly updating the foundation models would change the embedding distribution during training,  potentially disrupting gene program-based prototype initialization and weakening the biological structure that BP-Booster is intended to enforce. While joint fine-tuning may be beneficial in large-scale settings, it is not suitable for the small number of sample conditions we study and is therefore not adopted in this work.

---

> > > ### Author Response · Authors · 2025-11-24
> > > **Q4-3: How sensitive is BP-Booster to the number of prototypes and prototype dimension? Is there a risk of overparameterization in smaller datasets?**
> > >
> > > A4-3: According to the results in Table R3, adjusting the overlap threshold changes the number of prototypes and directly affects downstream performance. An overlap threshold of ≥10 is empirically supported by the ablation studies, which show that lower thresholds introduce noisy pathways dominated by single SVGs, while higher thresholds discard too much prior knowledge. This confirms the validity of the chosen threshold range.
> > >
> > > We also conducted ablation studies on the prototype dimensionality for the top 50 HVG prediction task, as shown in Table R6. The results indicate that different cancer types exhibit different performance trends under Virchow 2 when the prototype dimension changes. For selecting the default prototype dimension, we consider all datasets for each foundation model to determine a balanced setting.
> > >
> > > **Table R6: Performance comparison of different prototype dimensions on the SKCM and LUAD datasets for the top 50 HVGs prediction task (using Virchow 2)**
> > >
> > > | Dimension of Prototype | SKCM   | LUAD   |
> > > |------------------------|--------|--------|
> > > | 128                    | 0.6634 | 0.5959 |
> > > | 256                    | 0.6562 | 0.6004 |
> > > | 512                    | 0.6293 | 0.5794 |

---

> > > > ### Author Response · Authors · 2025-11-24
> > > > **Q4-4: My deepest concern towards this method is the positioning. Can you elaborate your method's contributions under the context of the concepts mentioned in the section of weaknesses? If not relevant, why the practice is unique? If yes, why existing methods cannot solve the issue?**
> > > >
> > > > A4-4: BP-Booster may superficially resemble feature transformation approaches such as domain adaptation, multimodal alignment, and feature recalibration, but it differs fundamentally in both concept and methodology. Domain adaptation focuses on reducing distributional shifts between domains, while multi-modal alignment methods typically learn a shared, bidirectional embedding space for retrieval or generation. Feature recalibration approaches reweigh feature channels based on internal feature statistics.
> > > >
> > > > In contrast, BP-Booster performs unidirectional, semantically driven refinement on frozen image embeddings from a foundation model. It explicitly reconstructs embeddings using biology-informed gene program prototypes and jointly optimises reconstruction and regression objectives, ensuring that the refined embeddings are both effective for gene expression prediction and interpretable through biological information. The core innovations of BP-Booster are: (1) prototype initialization using biological prior knowledge, (2) a reconstruction-plus-regression objective that enforces biological semantics, and (3) operating entirely on frozen embeddings to maintain stability under extremely limited data, these capabilities not achievable with previous domain adaptation, alignment, or recalibration approaches. With these three innovations, BP-Booster consistently outperforms SOTA methods across metrics, while also improve biological interpretability , an essential features for the applications of spatial transcriptomics in digital pathology. The interpretability is achieved through: 1) linking prototypes to known pathways, 2) semantic grounding through gene program-guided initialization with biologically informed priors (preserves biologically meaningful semantic features while task-irrelevant information is removed).

---

### Official Review · Reviewer_HyG1 · 2025-10-31

**Soundness:** 2
**Presentation:** 3
**Contribution:** 3
**Rating:** 4
**Confidence:** 4

**Summary:**

This paper proposes BP-Booster, a small module that can be plugged in between a frozen pathology foundation model and a downstream regressor for spatial transcriptomics gene‑expression prediction from H&E images. BP‑Booster uses prototype‑guided cross‑attention (prototypes query image embeddings) and gating mechanisms to shape a refined latent representation. Prototypes are either randomly initialized or initialized with gene programs built by intersecting spatially variable genes (SVGs) with MSigDB pathways, then aggregating expression across spatial locations. The experiments show significant performance improvement on Hest-1k and demonstrate the practical value of this module.

**Strengths:**

1. The paper introduces a clear and modular idea: Using prototypes as queries over frozen embeddings and the auxiliary regression/reconstruction losses are simple and easy to implement.
2. The cross‑model analysis is helpful to readers choosing backbones and also demonstrating the proposed methods work for different backbones

**Weaknesses:**

1. While I understand the problem setting is frozen image embedding + Ridge regression to predict the expression, I’m concerned the reported gains may largely come from additional params introduced by PGCA rather than from biological priors or prototype priors. Ridge cannot model nonlinearity, so a fair ablation should replace ridge with a parameter-matched MLP adaptor (same input/output dims and similar parameter count/FLOPs as PGCA) to isolate the effect of attention vs. capacity.
2. Improvements over a PCA adaptor are modest (Table 3) and not particularly compelling on their own.
3. Gene-set selection is under discussed. Relying only on MSigDB leaves some questions: gene sets vary widely in size and scope; what’s the principled selection strategy? With a ≥10-gene overlap threshold against SVGs, many pathways may pass by chance. For ST specifically, panels are often hand-curated, how does the method account for that?

**Questions:**

See my Weaknesses.

---

> ### Author Response · Authors · 2025-11-24
> **Q3-1: While I understand the problem setting is frozen image embedding + Ridge regression to predict the expression, I’m concerned the reported gains may largely come from additional params introduced by PGCA rather than from biological priors or prototype priors. Ridge cannot model nonlinearity, so a fair ablation should replace ridge with a parameter-matched MLP adaptor (same input/output dims and similar parameter count/FLOPs as PGCA) to isolate the effect of attention vs. capacity.**
>
> A3-1: To isolate the contribution of biological information (including both biological priors and biological supervision), we replace BP-Booster with a simple MLP-based gated autoencoder. This model uses the same decoder and regression head as BP-Booster, and its encoder contains the same number of parameters as PGCA. We conduct an ablation study on the SKCM dataset for the top 50 HVG prediction task. As shown in Table R5, BP-Booster consistently outperforms the gated autoencoder, demonstrating that the performance gains stem from the incorporation of biological information rather than from additional model parameters.
>
> **Table R5: Comparison among a simple MLP-based autoencoder (with and without biological supervision) and BP-Booster on the top 50 HVG prediction task in the SKCM dataset (Virchow 2 as foundation model)**
>
> | Method                                                      | Top 50 HVGs |
> |-------------------------------------------------------------|-------------|
> | Gated Autoencoder + reconstruction loss                     | 0.5973      |
> | Gated Autoencoder + reconstruction + regression loss        | 0.6448      |
> | BP-Booster                                                  | **0.6562**      |

---

> > ### Author Response · Authors · 2025-11-24
> > **Q3-2: Improvements over a PCA adaptor are modest (Table 3) and not particularly compelling on their own.**
> >
> > A3-2: Although the numerical improvements over PCA may appear modest, PCA is an exceptionally strong baseline in our small number of samples and parameter frozen embedding setting, where dominant linear components already explain most of the variance. BP-Booster still consistently outperforms PCA with a very small parameter budget, indicating that biologically guided prototypes provide addit ional, task-specific refinement beyond linear adaptors. Moreover, BP-Booster enables biological interpretability and the integration of prior knowledge, which PCA does not offer. BP-Booster leverages biologically informed prototypes to guide latent space reconstruction, which helps preserve relevant visual information about gene expression while simultaneously filtering out noise and redundancy—a limitation observed in non-guided methods like Principal Component Analysis (PCA).

---

> > > ### Author Response · Authors · 2025-11-24
> > > **Q3-3: Gene-set selection is under discussed. Relying only on MSigDB leaves some questions: gene sets vary widely in size and scope; what’s the principled selection strategy? With a ≥10-gene overlap threshold against SVGs, many pathways may pass by chance. For ST specifically, panels are often hand-curated, how does the method account for that?**
> > >
> > > A3-3: Our gene-set selection is not arbitrary. We use the MSigDB oncogenic and canonical pathway collections because they are curated, experimentally validated, and widely applied in cancer transcriptomics, ensuring biological specificity and avoiding heterogeneous gene lists. Importantly, gene program initialization is data-driven. Each pathway is filtered based on SVGs from the samples, which reduces random matches and anchors prototypes to tissue-specific expression patterns. The ≥10 overlap threshold is also empirically supported by the ablation studies (Refer to Table R3), which show that lower thresholds introduce noisy pathways dominated by single SVGs, while higher thresholds remove too much prior knowledge, confirming that our chosen range is reasonable. Finally, the method is not tied to MSigDB. Any hand-curated pathway resource or ST gene panel can be used instead, and the same SVG-intersection procedure will automatically adapt prototypes to the target dataset.

---

### Official Review · Reviewer_HSYj · 2025-11-01

**Soundness:** 2
**Presentation:** 2
**Contribution:** 1
**Rating:** 4
**Confidence:** 3

**Summary:**

The paper introduces Biology-Guided Prototype Booster (BP-Booster), a method designed to optimize H&E image embeddings from foundation models for improved task-specific adaptability in gene expression prediction. BP-Booster is based on a Q-former architecture, with a prototype initialization strategy proposed for the Q-former. Experimental results demonstrate that BP-Booster achieves better Pearson correlation coefficient (PCC) and mean squared error (MSE) performance compared to selected baselines, using different foundation models (UNI, CONCH, and Virchow2) as feature extractors.

**Strengths:**

- The paper focuses on an important problem, adapting representations from pre-trained H&E image foundation models for gene expression prediction.
- The paper shows better performance than the selected methods on the evaluated datasets.

**Weaknesses:**

- Unclear contribution

Adapting or fine-tuning parameters of foundation models typically enhances their task-specific performance. In the proposed BP-Booster, learnable parameters are added on top of the frozen foundation models to improve adaptability; however, the extent of this contribution to the overall task performance is not clearly analyzed. The method is closely related to parameter-efficient fine-tuning (PEFT), or even direct fine-tuning, but the paper does not provide a detailed study or comparison with existing PEFT approaches. Although the paper demonstrates better performance than the selected methods on the evaluated datasets, it does not clearly evaluate how the proposed method improves existing related approaches.

- Unclear setting:

There exist large-scale datasets collection such as STimage-1K4M and HEST-1K. It is unclear why the proposed method was not evaluated on these broader datasets to better demonstrate its generalization capability.

Based on the setup in Table 3, it is difficult to compare the method performance with and without the GP initialization, as the comparison is not clearly isolated.

**Questions:**

In Eq. 12, is any weighting applied to the reconstruction loss?

---

> ### Author Response · Authors · 2025-11-24
> **Q2-1: The method is closely related to parameter-efficient fine-tuning (PEFT), or even direct fine-tuning, but the paper does not provide a detailed study or comparison with existing PEFT approaches**
>
> A2-1: We clarify that BP-Booster is not a parameter-efficient fine-tuning (PEFT) method and does not adapt or modify the weights of the foundation model. BP-Booster operates solely on frozen image embeddings, with all learnable parameters external to the backbone. In contrast to PEFT methods (e.g., LoRA (Hu etal. 2022)), which aim to update internal model parameters in a parameter-efficient way, BP-Booster has a different objective: refining and reconstructing the embedding space by incorporating biological information and suppressing task-irrelevant visual information.  The performance gain arises not from fine-tuning the foundation model but from (i) gene program–based prototype initialization, (ii) prototype-guided cross-attention module, and (iii) the joint reconstruction–regression optimization. Our ablations in the paper and rebuttal focus on the core module (e.g., prototype count, initialization strategy, regression loss). Furthermore, using the same number of parameters in a simple MLP-based gated autoencoder (as shown in Table R5) fails to match BP-Booster’s performance, confirming that the improvement is structural rather than due to model size. In addition, PEFT approaches also generally require sufficient data to adapt parameters effectively, whereas BP-Booster improves embedding reconstruction and downstream task performance even in extremely data-limited scenarios, for examples patient data from the currently high-cost spatial transcriptomics technologies.
>
> For these reasons, we did not include empirical comparisons with existing PEFT methods. We will revise the paper to explicitly clarify these conceptual differences and explain why direct comparison is not appropriate in this context.
>
> Reference:
>
> Hu, E. J., Shen, Y., Wallis, P., Allen-Zhu, Z., Li, Y., Wang, S., ... & Chen, W. (2022). Lora: Low-rank adaptation of large language models. ICLR, 1(2), 3

---

> > ### Author Response · Authors · 2025-11-24
> > **Q2-2: There exist large-scale datasets collection such as STimage-1K4M and HEST-1K. It is unclear why the proposed method was not evaluated on these broader datasets to better demonstrate its generalization capability.**
> >
> > A2-2: We indeed utilized the HEST 1k dataset and selected multiple cancer-type sub-datasets following the settings described in the original paper. Our objective is to more effectively reconstruct image embeddings under extremely limited data conditions, thereby improving performance in downstream gene expression prediction tasks. Additionally, we also assessed datasets of different sizes within the HEST 1k database, including PRAD dataset contains 23 patient samples, representing a relatively large dataset in this database. Our method continues to perform well under this setting, demonstrating strong generalization ability.

---

> > > ### Author Response · Authors · 2025-11-24
> > > **Q2-3: Based on the setup in Table 3, it is difficult to compare the method performance with and without the GP initialization, as the comparison is not clearly isolated.**
> > >
> > > A2-3: In Table 3, each column corresponds to a different comparison method. The first column (ridge) serves as our baseline, where image embeddings from the foundation model are directly used to train a ridge regression model. The second column (ridge + PCA) applies PCA to reduce the dimensionality of the embeddings before training the ridge model. The third and fourth columns use BP-Booster with random initialization and gene program initialization, respectively, to reconstruct the image embeddings prior to ridge training. We will improve the clarity and presentation of Table 3 in the revised version.

---

> > > > ### Author Response · Authors · 2025-11-24
> > > > **Q2-4: In Eq. 12, is any weighting applied to the reconstruction loss?**
> > > >
> > > > A2-4: Our approach focuses on reconstructing image embeddings from foundation models and enriching them with biological information. Accordingly, both the gene expression regression loss and the image reconstruction loss are equally important, and we therefore did not apply any weighting method to the reconstruction loss. Future work can introduce learnable weighting parameters for the contribution from image reconstruction and gene expression prediction

---

### Official Review · Reviewer_oiFx · 2025-11-01

**Soundness:** 2
**Presentation:** 2
**Contribution:** 2
**Rating:** 4
**Confidence:** 4

**Summary:**

This paper introduces the Biology-Guided Prototype Booster, a lightweight module designed to enhance general-purpose embeddings from pathology foundation models specifically for gene expression prediction. The module uses a novel prototype-guided cross-attention mechanism, where prototypes can be initialized using biological priors.

**Strengths:**

- The concept of a "Biology-Guided" prototype initialization is a nice contribution. Using Spatially Variable Genes and known biological pathways to create semantically meaningful, learnable prototypes is a novel method for injecting relevant prior knowledge into the model.
- The experimental validation is extensive and provides strong evidence for the method's generalizability. The authors demonstrate that BP-Booster consistently outperforms not only strong published baselines but also a strong PCA baseline across nine different foundation models.

**Weaknesses:**

- The paper's core biological contribution is the Gene Program-guided initialization. However, the results show that the simpler Random initialization is often competitive and, in some cases, even better (e.g., Virchow2 on PRAD in Table 2 , or CONCH on SKCM in Table 3). This seems to weaken the "Biology-Guided" claim. Can the authors elaborate on why a random initialization is so effective and sometimes superior?
- The methodology uses two different sets of genes: Spatially Variable Genes for the prototype initialization and Highly Variable Genes for the regression loss. The rationale for this split is not entirely clear. Why is one set better than other for initialization and training supervision? Have the authors tried to flip them or even find a common set?

**Questions:**

- The final prediction pipeline appears to be a two-stage process: 1) Train the BP-Booster to generate embeddings, and 2) Train a separate regression model on these frozen embeddings. Why was this approach chosen over an end-to-end solution? Have the authors tried an end-to-end approach?
- The Gene Program-Guided Initialization, which filters MSigDB pathways by intersecting them with spatially variable genes, is a critical step. The model's performance and the number of prototypes seem highly dependent on this filtering. How sensitive is the model to this intersection threshold, and how many prototypes did this process typically generate for the datasets (IDC, SKCM, etc.)? Do different disease types benefit from different number of prototypes? What do these prototypes mean/ comprise?

---

> ### Author Response · Authors · 2025-11-24
> **Q1-1: Main biological contribution is the gene program-guided initialisation. But the simpler random initialisation is often competitive, in some cases even better (e.g., Virchow2 on PRAD in Table 2 , or CONCH on SKCM in Table 3).  why a random initialization is so effective and sometimes superior?**
>
> A1-1: Our overall model, regardless of random or BP initialization, outperforms the base reference model by (Jaume et al.2024) and other SOTA methods in all cases. The performance gain arises not just from initialization, but through denoising the foundation-model embeddings via three main contributions: (i) gene program–based prototype initialization, (ii) prototype-guided cross-attention module, and (iii) the joint reconstruction–regression optimization. Among these, our main contribution is the prototype and cross-attention module. Below we discuss the comparisons between random and gene program initialization. We highlight that, although gene program initialization does not improve performance markedly, the main benefit from using this initialization approach is to provide interpretability by leveraging biologically informed priors. Also refer to A3-1 and A3-2 regarding additional experiments and discussion on the role of gene program initialization in improving embeddings and performance.
>
> Across Highly Variable Gene (HVG, Table 2), and High Mean, Highly Variable Gene (HMHVG, Table 3), and cancer-marker prediction (refer to our new analysis data, Table R1), our results show that the relative effectiveness of random initialization versus gene-program (GP) initialization depends primarily on three factors: (1) the biological coherence of the target gene set, (2) its alignment with the inductive biases encoded in each foundation model’s embedding space, and (3) the technical variance of the gene set due to technological limitations of the spatial technologies that contributes more to the total variance compared to biological variance.
>
> For HVG prediction, highly variable genes (HVGs) are often heterogeneous, including abundant structural genes, house-keeping enzymatic genes, stress-response genes, or those lowly-expressed genes that change markedly due to technical noise (e.g. dropout events). In some foundation models, such as UNI and CONCH, whose embeddings are weakly aligned with spatial or biological structure, random initialization provides greater flexibility, allowing the model to discover task-specific patterns directly from the data. However, other foundation models clearly show GP initialization is better than random baseline, because their embeddings capture global or local structural patterns, semantic features, or spatial correlations that are weakly aligned with the SVG/pathway-based gene programs. In these cases, GP initialization helps guide the model toward meaningful biological variation, even for loosely structured HVGs.
>
> In the HMHVGs task, the target genes exhibit high mean expression and strong spatial variation, reflecting coordinated biological processes. The HMHVGs are therefore less random compared to HVGs and are more aligned with the SVG- and pathway-based gene programs used by GP initialization. As a result, models whose embeddings encode biological or spatial semantics benefit from GP, whereas morphology-dominant models such as Virchow 2 show the opposite trend: GP introduces a mismatched prior, making random initialization more effective by avoiding inductive-bias conflict.
> To further test whether biological relevance is the key factor, we evaluated cancer-marker prediction, where genes are tightly linked to tumour phenotype and microenvironment states. As expected, models with biologically aligned representations (UNI, Virchow2) clearly benefit from GP initialization. Only CONCH whose representations are strongly influenced by text-aligned histopathological semantics rather than biological pathway structure, which shows better performance with random initialization. These results reinforce the central conclusion: Random initialization outperforms GP only when the target gene set lacks biological coherence or when the model’s embedding space is poorly aligned with GP priors. When the genes are biologically meaningful (e.g., HMHVG, cancer markers), GP is consistently the better choice.
>
> Overall, these findings show that the cases where random initialization outperforms GP are not contradictory but follow directly from the interaction between biological prior suitability, task structure, and the inductive biases of foundation model embeddings.
>
> **Table R1. Comparison with methods on the SKCM cancer marker prediction task**
>
> | Method      | Random Initialization | Gene Program Initialization |
> |-------------|-----------------------|-----------------------------|
> | UNI         | 0.861                 | **0.8663**                      |
> | CONCH       | **0.8686**                | 0.8583                      |
> | Virchow 2   | 0.8342                | **0.8518**                      |
>
>
> Reference:
> Jaume, G., Doucet, P., Song, A., Lu, M. Y., Almagro Pérez, C., Wagner, S., ... & Mahmood, F. (2024). Hest-1k: A dataset for spatial transcriptomics and histology image analysis. Advances in Neural Information Processing Systems, 37, 53798-53833

---

> > ### Author Response · Authors · 2025-11-24
> > **Q1-2: The methodology uses two different sets of genes: Spatially Variable Genes for the prototype initialization and Highly Variable Genes for the regression loss. The rationale for this split is not entirely clear. Why is one set better than other for initialization and training supervision? Have the authors tried to flip them or even find a common set?**
> >
> > A1-2: The rationale for using two distinct gene sets (SVGs and HVGs) stems from the different roles of prototype initialization and supervised regression. Using SVGs to construct gene programs because they provide prototypes with spatially coherent and biologically meaningful priors, aligning well with the morphological patterns captured in H&E image embeddings. In contrast, not all HVGs carry such spatial information (as discussed in the A1-1). Moreover, HVGs are used for supervised regression because their high expression variability generates clearer gradients, offering stronger supervision signals. As suggested by the reviewer, to validate this design, we conducted additional experiments by flipping the roles of SVGs and HVGs on the SKCM dataset for HVG and cancer-marker prediction (which are shown in Table R2). The results show that using SVGs for prototype initialization and HVGs for regression consistently outperforms the flipped configuration.
> >
> > **Table R2: Comparison of two different strategies across two prediction tasks on the SKCM dataset (Virchow 2 as foundation model)**
> >
> > | Task            | SVG init + HVG regression | HVG init + SVG regression |
> > |-----------------|---------------------------|---------------------------|
> > | Top 50 HVGs     | **0.6562**                    | 0.6510                    |
> > | Cancer Marker   | **0.8518**                    | 0.8342                    |

---

> > > ### Author Response · Authors · 2025-11-24
> > > **Q1-3: The final prediction pipeline appears to be a two-stage process. Why was this approach chosen over an end-to-end solution? Have the authors tried an end-to-end approach?**
> > >
> > > A1-3: The pipeline is designed as a two-stage process because BP-Booster is trained end-to-end to refine the original image embeddings from foundation models, enriching them with biological information. The regression head in BP-Booster, which predicts the top 200 HVGs, is not intended as the final task but serves as a dense auxiliary supervision signal to guide the embedding toward biologically meaningful representations. Downstream tasks (such as predicting the top 50 HVGs, HMHVGs, PAM50 genes, and cancer markers) are trained on these refined embeddings. Attempting to combine these stages into a single end-to-end approach could introduce conflicting gradients, potentially degrading model performance, for example in the scenarios that the downstream prediction of genes with high technical noise and low biological variation.

---

> > > > ### Author Response · Authors · 2025-11-24
> > > > **Q1-4: How sensitive is the model to the threshold of filtering MSigDB pathways by intersecting them with spatially variable genes, and how many prototypes did this process typically generate for the datasets (IDC, SKCM, etc.)? Do different disease types benefit from different number of prototypes? What do these prototypes mean/ comprise?**
> > > >
> > > > A1-4: We perform ablation studies to investigate how model performance and the number of prototypes is affected by varying the threshold used in gene program initialization, as shown in Table R3. When the threshold is set to 1, meaning a gene program is activated if it shares at least one SVG with the selected genes, a total of 64 gene programs is included. However, this leads to decreased performance compared to a threshold of 10. The likely reason is that some activated gene programs contain only a single SVG, thus introducing noise during initialization rather than providing useful prior knowledge. Conversely, when we increase the threshold to 15, only two gene programs remain after filtering. In this case, the prior information becomes overly limited, offering insufficient biological guidance and causing a slight drop in model performance. These observations demonstrate that the threshold plays a critical role in balancing the richness and noise level of prior knowledge incorporated into prototype initialization.
> > > >
> > > > **Table R3: Ablation study on prototype number sensitivity by varying the threshold in gene program initialization on the SKCM dataset (Virchow 2 as foundation model).**
> > > > | Threshold | Number of Prototypes | Top 50 HVGs |
> > > > |-----------|-----------------------|-------------|
> > > > | 1         | 64                    | 0.6471      |
> > > > | 10        | 12                    | **0.6562**      |
> > > > | 15        | 2                     | 0.6548      |
> > > >
> > > > For different disease types, the gene programs differ. We extract gene programs list for SKCM and LUAD datasets, which is shown in Table R4. The table shows different cancer type has different number and type of pathway selected for prototype initialization. Lung cancer is dominated by classical oncogenic signalling, including EGFR, ALK, KRAS–RAF–MEK, and AKT–mTOR, leading to enrichment of genes in RTK activation and cell-cycle progression(Lynch etal. 2004, Drosten etal. 2020). In contrast, skin cancer, especially melanoma, shows more of stemness and developmental pathways such as Wnt/LEF1 and SHH, together with RAF activation, reflecting more genes in dedifferentiated and MAPK-driven profiles (Davies etal. 2002).
> > > >
> > > > **Table R4: Gene program lists for the SKCM and LUAD datasets derived from MSigDB pathways**
> > > >
> > > > | SKCM                                | LUAD                          |
> > > > |-------------------------------------|-------------------------------|
> > > > | P53 DN.V1 DOWN                      | HOXA9 DN.V1 UP                |
> > > > | LEF1 UP.V1 UP                       | MEK UP.V1 UP                  |
> > > > | SNF5 DN.V1 UP                       | RPS14 DN.V1 UP                |
> > > > | ESC V6.5 UP EARLY.V1 DOWN           | EGFR UP.V1 UP                 |
> > > > | CAMP UP.V1 DN                       | ALK DN.V1 UP                  |
> > > > | RAF UP.V1 DOWN                      | AKT UP MTOR DN.V1 DOWN        |
> > > > | P53 DN.V1 UP                        |                               |
> > > > | NFE2L2.V2                           |                               |
> > > > | BMI1 DN MEL18 DN.V1 UP              |                               |
> > > > | BMI1 DN.V1 UP                       |                               |
> > > > | STK33 SKM UP                        |                               |
> > > > | GCNP SHH UP EARLY.V1 DOWN           |                               |
> > > >
> > > > Reference:
> > > >
> > > > Lynch, T. J., Bell, D. W., Sordella, R., Gurubhagavatula, S., Okimoto, R. A., Brannigan, B. W., ... & Haber, D. A. (2004). Activating mutations in the epidermal growth factor receptor underlying responsiveness of non–small-cell lung cancer to gefitinib. New England Journal of Medicine, 350(21), 2129-2139.
> > > >
> > > > Drosten, M., & Barbacid, M. (2020). Targeting the MAPK pathway in KRAS-driven tumors. Cancer cell, 37(4), 543-550.
> > > >
> > > > Davies, H., Bignell, G. R., Cox, C., Stephens, P., Edkins, S., Clegg, S., ... & Futreal, P. A. (2002). Mutations of the BRAF gene in human cancer. Nature, 417(6892), 949-954.

---

### Author Response · Authors · 2025-11-24

We sincerely thank all reviewers for their thorough evaluation of our manuscript and for the constructive feedback and insightful suggestions provided. We greatly appreciate the time and effort each reviewer has devoted, which has substantially improved the quality and clarity of our work. We have carefully addressed every comment in our point-by-point response and incorporated all necessary revisions into the updated version submitted before 3 December.

---

### Meta-Review · Area_Chair_NGxp · 2026-01-10

**Summary:**

Three reviewers rated the paper as marginally below the acceptance threshold. They acknowledged the general motivation of incorporating prior biological knowledge for this challenging task, but raised several concerns, including insufficiently grounded claims of contribution, limited justification of the proposed methodology and experimental setting, and marginal performance improvements over existing methods. The authors provided rebuttals addressing these concerns. AC believes that some issues related to the methodological design and experimental setup could potentially be resolved through clarification or revision. However, it is difficult to envision that the core concerns regarding the limited performance gains and the strength of the claimed contributions could be fully addressed within the current framework. Reviewer N92h recommended rejection, noting that the proposed approach appears closely related to existing techniques in domain adaptation, multimodal alignment, and feature recalibration, yet the paper does not sufficiently discuss or position itself with respect to these lines of work. While AC believes that the rebuttal could mitigate this concern, a more thorough discussion of these related methods would still be necessary and beneficial. Overall, AC finds that the paper has merit in its motivation and problem formulation, but believes that the technical development, methodological grounding, and experimental validation require further strengthening before the work would meet the bar for acceptance at a top venue. Therefore, the final recommendation is Reject.

**Reviewer Concerns:**

Reviewer oiFx raised two major concerns regarding the gene program–guided initialization and the rationale behind splitting genes into two sets. In their rebuttal, the authors demonstrated that performance is indeed sensitive to initialization, and that the proposed initialization strategy may even be inferior to random initialization under certain data distributions. However, the paper does not provide an explicit or principled justification for the proposed gene split. As a result, the AC believes that the primary concerns raised by Reviewer oiFx are likely to remain unresolved.

Reviewer HSYj questioned the relationship between the proposed method and parameter-efficient fine-tuning (PEFT), or even direct fine-tuning, noting the lack of comparison with existing PEFT approaches. The authors clarified that BP-adaptor does not aim to modify model parameters, but instead refines and reconstructs the embedding space by incorporating biological priors while suppressing task-irrelevant visual information. AC finds this explanation largely reasonable and believes that this concern could be mostly addressed. However, a clearer and more explicit rationale for why fine-tuning is not considered or preferred should be included in the main paper. Reviewer HSYj also noted the limited evaluation of generalization on large-scale datasets. While the authors cited experiments on the HEST-1k dataset, this may still fall short of fully meeting the reviewer’s expectations in terms of evaluation scope.

Reviewer HyG1 suggested a fair ablation by replacing ridge regression with a parameter-matched MLP adaptor. The authors conducted this experiment, showing that ridge regression performs competitively. However, Reviewer HyG1 also pointed out that the improvements over a PCA-based adaptor are modest and not particularly compelling. AC agrees that the rebuttal does not sufficiently strengthen the argument on this point.

Reviewer N92h observed that the proposed approach appears closely related to existing techniques in domain adaptation, multimodal alignment, and feature recalibration. The authors responded that while BP-Booster may superficially resemble feature transformation methods, it differs fundamentally in both concept and methodology. AC agrees that this concern can be sufficiently addressed with clearer positioning and discussion.

**Reviewer Scores:**

Reviewer oiFx is likely to maintain the original score (Marginally below the acceptance threshold), as the primary concerns appear to remain unresolved. Reviewers HSYj and HyG1 may keep their scores unchanged or, in a more optimistic scenario, increase them slightly. Reviewer N92h may potentially revise the score from Reject to Marginally below the acceptance threshold. Overall, while minor score adjustments are possible, it is difficult to envision any reviewer substantially changing their initial stance to strongly support acceptance.

---

### Decision · Program_Chairs · 2026-01-26

Reject